# Evaluation of a Numerical, Real-Time Ultrasound Imaging Model for the Prediction of Litter Size in Pregnant Sows, with Machine Learning

**DOI:** 10.3390/ani12151948

**Published:** 2022-07-31

**Authors:** Konstantinos Kousenidis, Georgios Kirtsanis, Efstathia Karageorgiou, Dimitrios Tsiokos

**Affiliations:** 1Department of Agriculture, School of Geosciences, International Hellenic University, Sindos, 574 00 Thessaloniki, Greece; ekarageorg@admin.teithe.gr; 2Department of Electrical and Computer Engineering, School of Engineering, University of Thessaly, 382 21 Volos, Greece; gkirtsanis@uth.gr; 3Research Institute of Animal Science, Hellenic Agricultural Organization DEMETER, 581 00 Giannitsa, Greece; tsiokosd@gmail.com

**Keywords:** sows, pregnancy, ultrasonography, numerical model, artificial neural network, machine learning, litter size

## Abstract

**Simple Summary:**

The use of a numeric model to quantify real-time ultrasonographic (RTU) imaging is a prominent methodfor predicting the expected litter size by training an artificial neural network (ANN) to minimize the error of the prediction measured by metrics, such as root square mean error and mean absolute error. Time of the RTU application is a critical factor for such a prediction. Rated scale values (RSV) obtained from the RTU images relate to the accurate diagnosis of pregnancy and of litter size, suggesting the potential of a generalized use of the model in various farm conditions. Ultimately, the employment of the model in machine learning for an automated prediction of litter size can be used as a routine on-farm procedure for the efficient management of gestating sows.

**Abstract:**

The present study aimed to evaluate the accuracy of a numerical model, quantifying real-time ultrasonographic (RTU) images of pregnant sows, to predict litter size. The time of the test with the least error was also considered. A number of 4165 pregnancies in Farm 1 and 438 in Farm 2 were diagnosed twice, with the quality of the RTU images translated into rated-scale values (RSV1 and RSV2). When a deep neural network (DNN) was trained, the evaluation of the method showed that the prediction of litter size can be performed with little error. Root square mean error (RMSE) for training, validation with data from Farm 1, and testing on the data from Farm 2 were 0.91, 0.97, and 1.05, respectively. Corresponding mean absolute errors (MAE) were 2.27, 2.41, and 2.58. Time appeared to be a critical factor for the accuracy of the model. The smallest MAE was achieved when the RTU was performed at days 20–22. It is concluded that a numerical, RTU imaging model is a prominent predictor of litter size, when a DNN is used. Therefore, early routinely evaluated RTU images of pregnant sows can predict litter size, with machine learning, in an automated manner and provide a useful tool for the efficient management of pregnant sows.

## 1. Introduction

Since the application of artificial insemination (A.I.) is based on semen from A.I. stations, management of pig reproduction at a farm level is currently limited to the efficient reproductive function of the sow herd [1]. The systematic management of the sows focuses on the continuous manifestation of reproductive cycles, maximizing her output estimated as piglets weaned per sow per year [2], and minimizing the loss, expressed as non-productive (empty) days, due to failure of fertilization or pregnancy diagnosis and delayed onset of oestrus. The most common reason for keeping non-pregnant females in pig farm operations is the failure to identify sows that do not conceive soon after breeding [3].

Early diagnosis of pregnancy is provided, with the necessary information for the proper management of nutrition and housing for the gestating sow, in order to secure the successful completion of pregnancy and the farrowing of a large litter of well developed and uniform piglets. It is traditionally performed by a boar that identifies sows who do or do not return to oestrus. Alternative methods include measurement of urinary oestrogens, oestronesulphate assays, and ultrasound techniques [4]

Transabdominal real-time ultrasonography (RTU) is currently the most common method of pregnancy diagnosis in pigs, as it is non-invasive and can be performed as early as 21 days after mating with more sensitivity, specificity, and, moreover, with an efficiency close to 95% [5]. Pregnant sows imaging is based on the presence of fluid in embryonic vesicles as a concept and uses their increase in diameter with time, starting from 1.0 mm on day 9 post-mating and reaching 10 mm in diameter by days 18–22 [6]. RTU technology limited the time and significantly improved the accuracy of diagnosis, and also assisted in reducing non-productive days in pig production [7]. However, RTU imaging of pregnant sows is far from being fully exploited, as there is a lack of information regarding the precise day for the most accurate diagnosis. It is also suggested that reproduction research should be directed towards the production of predictable numbers of healthy, uniform piglets [4]. Combined with the use of farm data analysis, sows’ reproductive potential can be maximized and herd productivity can improve [8]. Such information can derive from the analysis of the RTU image of the pregnant sow along with the day of application and other potential factors that may affect the image, such as the age and breed of the sow. Furthermore, there is the potential of predicting the number of piglets, based on the general principle that an unclear and uncertain image is usually an indication of small litter size [9]. Estimation of the expected litter size by the thorough examination of transabdominal or transrectal RTU imagingwas studied in species such as the snowshoe hare, the dog, as well as sheep and goats [10,11,12,13]. In the pig, research on the prediction of litter size focuses on measuring maternal oestrone sulphate blood levels [14,15].

There are earlier published results that demonstrate the potential of predicting the expected litter size using an ultrasonic Doppler method [16] and more recently by transforming the quality of the transabdominal RTU image of pregnant sows into a numerical model [9]. Machine learning technology using an artificial neural network (ANN), significantly increased the value of such models to correspond to 10% mean expected deviation between the predicted and the actually obtained litter size [17]. 

Estimation of the expected litter size during early gestation is information highly beneficial for the nutritional, reproductive, and housing management of pregnant sows. The main questions to be answered in such a quest are the time of the RTU scan that produces the most valuable results, along with the applicability of such a model in a variety of conditions. The present study aims to answer both these questions, by establishing the correlation between the quantified evaluation of the RTU image at various post-insemination times and the expected litter size, with the use of artificial neural networks. Consequently, there can be a model produced for the automated prediction of litter size based on the RTU imaging through machine learning.

## 2. Materials and Methods

### 2.1. Experimental Design

The present study was designed in view of a thorough evaluation of the performance of teal-time ultrasonography (RTU) under commercial farm conditions. The aim of the study was to determine the best post-insemination interval for the application of RTU, as well as to investigate the predictive value of a quantified evaluation of the RTU image for the prognosis of the expected litter size in a different farm.

### 2.2. Farms, Animals, and Application of RTU

The study was conducted at two commercial pig farms in Greece. In Farm 1, collection of data were made for a period of seven years (October 2014 to September 2021). The housing capacity of the farm is 300 sows and the number of present sows during the experimentation period varied annually between 254 and 285 sows. The experimental sow population comprised of 1205 sows, among which 321 sows were F2 crossbreds (large white × landrace), 839 sows were F1 crossbreds (landrace × large white) and 45 sows were purebred large white (LW) nucleus sows. Parity number of all recorded sows varied between 1 and 10. The sows were housed in individual crates from weaning to the establishment of pregnancy. Oestrus was detected by the standingheat reflex after exposure to a boar’s presence. Fertilizations were performed entirely by artificial insemination with semen produced on-farm for all terminal sows and purchased semen from an AI station for the purebred nucleus sows. Double insemination was carried out in each oestrus with semen containing > 3 × 10^9^ spermatozoa. For the on-farm collected semen, six boars were used, whilst utilization of two different boars for double insemination was recorded as a separate case. Litter size was registered after the placental expulsion (completion of parturition). In very large or extremely small litter sizes, cross-fostering was then practiced.

Farm 2 was a 240-sow unit, and data were collected for a period of 10 months (January to October 2021). The experimental sow population comprised of 278 sows, of which 264 were F1 sows and 14 were purebred LW nucleus sows. Parity number varied between 1 and 9. Farming conditions were generally similar to those described for Farm 1. In Farm 2 all inseminations were made using single-sire semen from an A.I. station.

RTU pregnancy diagnosis was performed transabdominally, twice on each inseminated sow, with an average interval of 15 days between the two scans. The interval from insemination to first RTU scan varied in Farm 1 from 16 to 61 days post-insemination, whereas for the second RTU scan, the interval varied between 26 and 77 days post-insemination. In Farm 2, the intervals from insemination to first and second RTU scans varied from 20 to 44 days and 32 to 62 days, respectively. 

In Farm 1, a real-time ultrasound scanner (Agroscan A7, Echo Control Medical, 16000 Angoulême, France) provided with a 5 MHz linear transducer and set at a depth of 13 cm was used for the first 1010 cases. It was then replaced by an updated device (ImaGo.S, Echo Control Medical, 16000 Angoulême, France) using the same settings. An apparatus similar to the latter was used in Farm 2. For the operation of the diagnosis, the transducer was held against the hind abdomen at an angle of 45°, and exterior to the second nipple pair line. Mineral coupling gel was also utilized for the effective transmission of the ultrasound waves, as suggested by Knox and Althouse [9,18]. The scan was performed by the same operator in both farms.

The RTU pregnancy diagnosis was not limited to a ‘yes or no’ verdict. In each scan, the image retrieved upon scanning was subjectively evaluated and scored. For this purpose, a previously established 10-point rating scale was used, with objectively estimated values given for each RTU image. The rated scale value (RSV) was based on the level of coverage of the image with developing embryonic vesicles. Rated scale value “0” corresponded to a complete absence of pregnancy indications, and RSV “10” was given when the image was completely covered by numerous, well-developed embryonic vesicles [9], translated as a prominent indication of a largelitter pregnancy. According to this scale (Appendix A), a RSV (score) was given for each case, whilst half values were used when necessary.

### 2.3. Data Collection

During the experimentation period, 4165 matings in Farm 1 and 438 in Farm 2 led to farrowings and the results were included in the database. The data collected for the purpose of the present study were based on the recording of the following parameters:Sow ID (ear tag number)Sow breed/lineSow parity numberInsemination date1st scan date1st scan score (RSV1)2nd scan date2nd scan score (RSV2)Farrowing dateLitter size (total pigs born).

Two extra parameters were also calculated, based on the collected data:Day 1 (as the interval from insemination date to 1st scan date)Day 2 (as the interval from insemination date to 2nd scan date)

The cases of sows that were not impregnated or did not complete pregnancy were excluded from the database, on which the artificial neural network was applied. There were also missing values in Farm 1, in either RSV1/Day 1 or RSV2/Day 2 due to the replacement of the apparatus and as a consequence of COVID-19 restrictions.

### 2.4. Data Analysis—Structure of the Artificial Neural Network

The goal of this research is to create an artificial model to predict the expected litter size based on six different input factors of the sow, and thus, a deep neural network (DNN) was trained on the given data, due to its ability to be trained on complex data. Data from two different farms were used for the needs of the training of the ANN. Data from Farm 1 were split into an 80% training dataset and 20% validation dataset, while the whole dataset collected from Farm 2 was used for testing of the model. Farm 1 and Farm 2 contained 4139 and 438 cases, respectively. The scaling of the input and output data were calculated with the mean and standard deviation of the training data, to assure that validation and testing data were not playing any significant role in the training dataset. The 6 input parameters were set to be: (a) parity number, (b) Day 1, (c) RSV1, (d) Day 2, (e) RSV2, (f) sow breed, based on a previous study [17], and the output was the expected prediction of litter size. The structure of the DNN is presented hereafter. The DNN consists of the (a) input layer, (b) hidden layer 1, (c) hidden layer 2, (d) hidden layer 3 and (e) output layer. Activation functions of the hidden layers are used to assist the training of the DNN. Among the available activation functions, relu [19] was used in the model for the better performance it provides. Furthermore, the dropout [20] technique was applied to the hidden layers due to its ability to avoid the model’s over-fitting on the training data. Adam optimizer [21] was used during the training of 100 epochs with a fixed 10 batch size. Ten (10) runs of experiments on the training of the model were conducted with different random initializations producing the average and standard deviation of the root square mean error (RMSE) loss function for training, validation, and testing datasets to plot the results. Finally, the best model was chosen to present the results of the mean and standard deviation absolute error according to Day 1 and Day 2.


**Layer Function**

**Shape**

**Parameters**

**Dropout**

**Activation**
Input Layer6---Hidden Layer 1644480.2reluHidden Layer 285200.2reluHidden Layer 3190.2reluOutput Layer1--
The number of total parameters was 977.

## 3. Results

The results of the present study are presented for the purpose of verifying the uniformity of the database between the two farms, as well as establishing the applicability and the predictive value of the numerical model for the RTU imaging of pregnant sows, with the use of an ANN. 

### 3.1. Descriptive Statistics

The parameters studied are presented in Table 1 as the overall results from Farm 1, and the fraction of the results from Farm 1 in the year 2021, which correspond in time with the results from Farm 2. These parameters are the RSV and day values, as well as litter size, parity number, and breed of the sow. 

Table 1 demonstrates that there is a statistically significant difference between the mean RSV2 from Farm 1 ’21 and Farm 2 and the actual litter size measured at birth. Both differences are in favor of Farm 1 ’21, indicating that higher RSV2 coincide with larger litters. However, this is not the case for the mean RSV1 values from Farm 1 ’21 and Farm 2, and the average parity number of the two sow populations. The standard deviation for the latter variable was also comparable between the two data sets (1.89 and 1.93, respectively), which suggests that the two populations are relatively uniform. 

In Table 2, data from the two RTU scans are portrayed, grouped in weekly time intervals according to the routine weekly examination that is performed on-farm. 

Based on the results from Table 2, it is becoming obvious that the earliest time with the highest RSV is in days 29–35, post-insemination. This is true for RSV1 in all farm data sets and is confirmed by RSV2 measurements of the first groups of RSV1 (15–21 days), which have a mean Day 2 value of 36–37. This observation strengthens the hypothesis that the most indicative scores for pregnancy diagnosis are obtained in week 5, post-insemination, coupled by equal or higher RSV2 scores after day 52. In terms of litter size, there is a trend among the groups of decreasing sizes, which appears to stabilize in the middle, most populated groups, at levels close to the mean values of the whole populations in both farms. It is, therefore, indicated that in both farms, the greatest litter size values are obtained in the first group, although RSV1 has the lowest scores due to the very early RTU examination. These findings suggest that, although the best RTU images may give a positive indication of pregnancy, high RSV values do not consistently coincide with large litter sizes.

### 3.2. Artificial Neural Network Application

#### 3.2.1. Dataset

In order to exploit the information contained in the data base, an ANN was employed for the thorough investigation of the potential of the numerical model to predict litter size in an accurate and replicable way. For this purpose, data from Farm 1 were used for the training and validation of the ANN and the model was tested on data from Farm 2. In Figure 1 and Figure 2 the distribution of the data from Farms 1 and 2 are presented, according to RSV1 and 2, the litter size, and sows’ parity number. For the purpose of presentation and balanced distribution of cases in groups, RSV was grouped into five groups: 1–6, 6 ≤ 7, 7 ≤ 8, 8 ≤ 9, and 9 ≤ 10. In the same respect, litter size was grouped as follows: 1–8, 9–11, 12–14, 15–17, and 18 or more, whilst only parities of 7 or more were utilized in this grouping. The charts are retrieved from the ANN in order to represent the exact data on which the network was based.

It is becoming apparent from Figure 1 and Figure 2 that, although RSV1 and 2, litter size and parity number varied between farms, and data distribution in the RSVs was similar. The difference in litter size in favor of Farm 2 seems to be due to the highest contribution of litters of 15–17 piglets, instead of the group of 12–14 piglets on Farm 1. In terms of parity number, less sows of parity 4 and 7 or more appear to be the reason why lower mean values are present in Table 1. Sow breed was also included in the ANN, but the distribution of breed is not portrayed in Figure 1 and Figure 2, as it is described in Table 1.

#### 3.2.2. ANN Results on the Prediction of Litter Size

The progress of the RMS values over 100 epochs, obtained by the employed ANN, along with their standard deviations are illustrated in Figure 3. In charts (a), (b), and (c) the RMSE and s.d. for training, validation, and testing are presented. Chart d. contains all three series under the same scale for the purpose of comparison. 

In Figure 3 it is demonstrated that the RMSE values obtained from the application of the ANN are relatively low, indicating that the model can produce predicted litter size values with little error. This in turn suggests that the prediction of the expected litter size based on the score (RSV) of the RTU imaging can be performed in a relatively accurate way. In a more detailed interpretation of the results, through 100 epochs, training RMSE (0.914) is continuously declining and there is substantial evidence that it could further improve if more epochs were adopted under the model. However, this is not the case for validation (RMSE: 0.968), where no further improvement, if not a slight rise of RMSE values, is apparent. The highest value is met in the testing series (RMSE: 1.052); an observation which could be expected, as the training and validation datasets come from Farm 1, whereas the testing dataset is based on observations from Farm 2. 

For the further evaluation of the quality of predictions achieved by the ANN, the mean absolute error (MAE), along with its standard deviation for training, validation, and testing are presented in Table 3.

MAE results from the ANN confirm those of RMSE in terms of absolute values and comparison among the training dataset, validation dataset, and testing. The overall values are higher than the equivalent RMSEs, which is a reasonable observation based on the mathematical equation of each metric. Therefore, the MAE and s.d. (standard deviation) values are not far from what was expected, further confirming the potential of the ANN to produce reliable predictions of litter size. 

To finally investigate which time of application of the RTU scan produces the most reliable predictions, the testing dataset from Farm 2 was divided in groups based on the days in pregnancy that the two scans were performed, namely Day 1 and Day 2. The grouping was generally based on 3-day intervals, with the marginal groups containing more days if necessary, for the purpose of avoiding a small number of observations in a group. The group division for Day 1 and Day 2 can be viewed in Table 4 and Figure 4 (a and b, respectively). The corresponding MAE and s.d. obtained from the separate testing on each group are presented as values in Table 4, and more graphically in Figure 4.

The results presented in Table 4 and Figure 4 indicate that the prediction of litter size with minimal error is achieved by the RTU image evaluations obtained as early as days 20–22 (Day 1). This is coupled by the lowest MAE values in days 41–43 and 44–46 in the Day 2 evaluations. In addition, where Day 1 and Day 2 overlap, similar MAE values are obtained, indicating that the results are consistent and repetitive. Finally, a general observation from the results that can also be valuable is that, in both RTU image evaluations performed on Day 1 and Day 2, the fluctuation of the MAE values becomes more moderate with time. 

## 4. Discussion

The initial descriptive statistics from the present study demonstrate that the data collected were retrieved from two well-managed farms, with good reproductive output from their sow herds. What is interesting is the fact that in Farm 1, where the data were collected since October 2014, there was a significant increase in litter size, possibly due to the genetic progress achieved [22]. This is becoming obvious from the difference in litter size between the Farm 1 and Farm 1 ’21, as it increased from 13.77 total piglets born to 15.26. This observation confirms the successful management of this farm, as it is close to what is achieved in the most successful breeders of hyperprolific sow lines, where an extra 2.4 piglets born per litter were attained since 2011 [23]. Another figure that supports that Farms 1 and 2 are representative of a modern pig enterprise is the mean parity number, which is close between Farm 1 ’21 and Farm 2 (3.25 and 3.22, respectively). This indicates that a good rate of sow replacement is practiced in both farms as breeding eligible gilts into the breeding herd is an important driver of sow lifetime productivity [24]. 

A valuable outcome from the results of this study is that an effort was made for the exploitation of all the available information from a routine on-farm procedure such as pregnancy diagnosis and related data. Farm data analysis can accurately monitor lifetime performance in individual sows [22], and the application of a numerical model for the RTU imaging of pregnant sows as a predicting factor for litter size is a valued practice for the successful management of sows [9]. In this respect, the numerical model for the quantification of the quality of RTU images of pregnant sows provided the necessary database. The results show that the highest mean RSV value, reported for RSV2 in Farm 1′21 (9.44), coincided with the largest mean litter size (15.26).

Time is a critical factor for the attained measurements. RSV increased with time, with the earliest high RSV1 observed in both farms at 29–35 days post-insemination, followed by a decline in both RSV1 and RSV2 until day 48 and a new increase after day 50. These findings are in agreement with the work of Miller et.al. [25], who found a similar pattern by measuring the uterine fluid diameter, reporting that fluid diameter increased to day 30, decreased to day 39, and increased thereafter. Although in the present study a delay in the increase in the RSV was evident, the similarity in the patterns is striking, as the quantification of the RTU image was based on the size and number of embryonic vesicles and the level of coverage of the image they produced. However, in the adapted numerical model, RSVs with values of 1 and higher imply, with an increasing certainty, that a pregnancy is established. Therefore, even the lowest values attained as early as 16–21 days were a positive diagnosis of pregnancy. These results are in agreement with the majority of the available literature, where it is broadly suggested that diagnosis of pregnancy, using Doppler ultrasound technology, can be made with high predictive value, specificity, and accuracy as early as on day 23 and up to day 30 [2,5,26,27,28,29]. Knox and Flowers [27] also suggested that, when a 5 MHz transducer was used, there was no technician difference in the accuracy of pregnancy diagnosis from day 21.

In the quest to predict litter size, based on the RSVs attained in the confines of the present study, the employed DNN produced very promising results. Many studies suggested the use of deep learning in the development of recommendation systems. DNNs take multiple criteria evaluations into account and can make the most of intricate, nonlinear interactions, and so, they can provide more accurate suggestions. The DNN output in root mean squared error (RMSE) and mean absolute error are metrics that show how accurate predictions are and the amount of deviation from the actual values [30,31]. As measures of the validity of the prediction model for the litter size based on the RSVs, they were only used once in a previous study [17]. The RMS and standard deviation were lower in the present study for the training dataset, validation dataset, and testing, probably due to the larger database used. MAE for testing was somewhat higher, a possible explanation being that the testing dataset was collected from a different farm. These results support the potential of a generalized use of the model in various farm conditions and ultimately its employment in machine learning for an automated prediction of litter size as a routine on-farm procedure [9,17]. The usefulness of such a prediction is highlighted by various researchers using RTU technology for counting the gestated litters in other species [10,11,12,13], or by measuring oestrone sulphate blood levels in sows [14,15]. However, the large litters in pigs are difficult to predict accurately by counting with ultrasonography, and the use of RTU for determining the potential numbers of piglets in a litter is not advised [27]. Earlier studies demonstrated the potential of roughly predicting litter size from RTU imaging, relating the quality of the image (number of black spots) with litter size [16,32,33]. Yet, never was there a report on the combination of the time and accuracy of such predictions. The results from the present study demonstrate that the prediction of litter size with minimal error is achieved by the enumerated RTU image obtained as early as days 20–22 (Day 1). RSV2 had the lowest MAE values in days 41–43 and 44–46. Both these results suggest that the prediction model performs better when embryonic vesicles are moderate in size. Hence, the hypothesis proposed is that the discrimination between larger and smaller litters is more accurately achieved at times when lower RSV values are recorded, which is in agreement with a model proposed previously [9].

## 5. Conclusions

The use of a numeric model to quantify RTU imaging is a prominent method for predicting the expected litter size by training an ANN to minimize the error of the prediction. Time of the RTU application is a critical factor for such a prediction and further research should be orientated towards the definition of the best time and training of the ANN model. In this way, the automated prediction of litter size by a simple RTU scan will provide the pig industry with a valuable tool for optimizing the management of gestating sows. 

## Figures and Tables

**Figure 1 animals-12-01948-f001:**
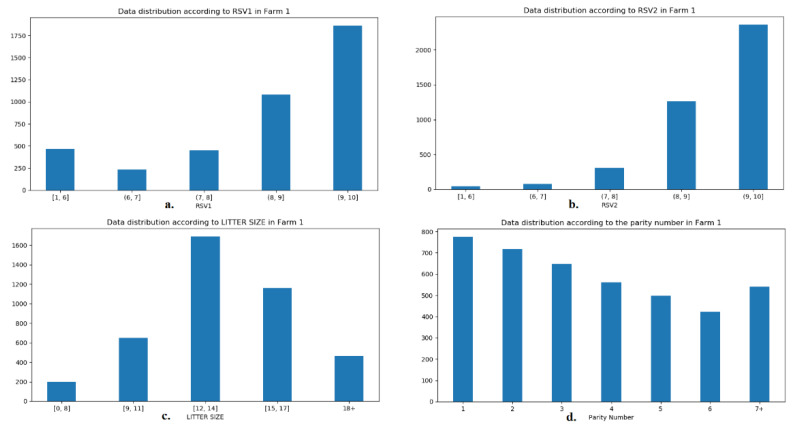
Distribution of the ANN data according to (**a**) RSV1, (**b**) RSV2, (**c**) litter size, and (**d**) parity number from Farm 1.

**Figure 2 animals-12-01948-f002:**
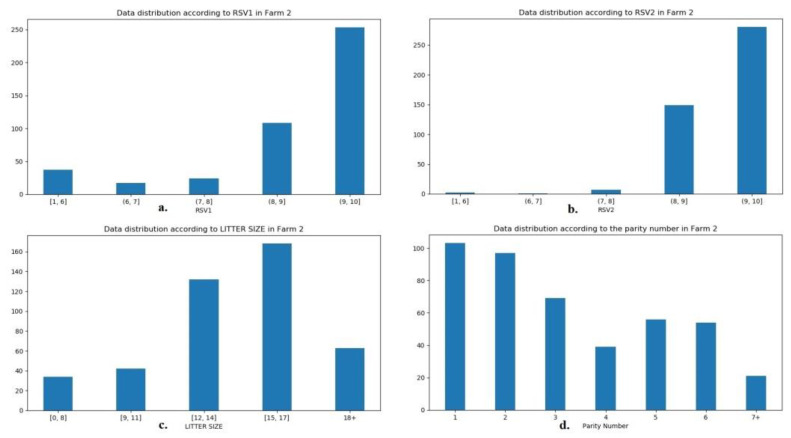
Distribution of the ANN data according to (**a**) RSV1, (**b**) RSV2, (**c**) litter size, and (**d**) parity number from Farm 2.

**Figure 3 animals-12-01948-f003:**
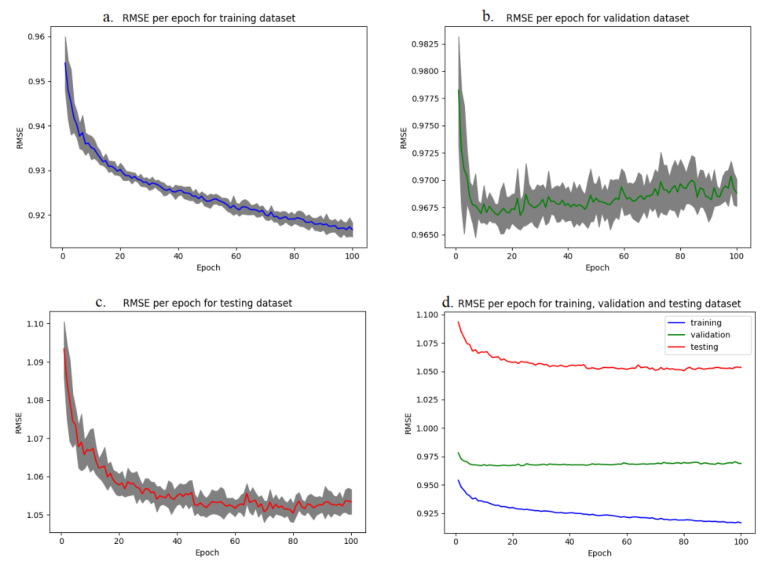
Metrics of the prediction model expressed in RMSE with s.d. for (**a**) training dataset, (**b**) validation dataset, (**c**) testing, and (**d**) training dataset, validation dataset, and testing under the same scale (without s.d.).

**Figure 4 animals-12-01948-f004:**
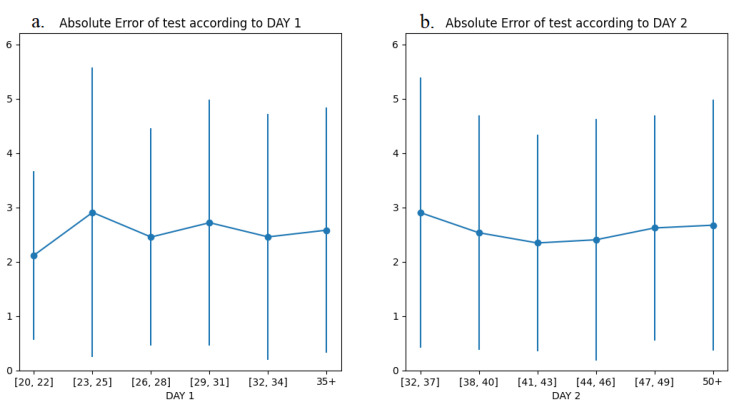
Testing MAE with s.d. for 3-day intervals’ groups in: (**a**) Day 1, (**b**) Day 2.

**Table 1 animals-12-01948-t001:** Mean RSV, litter size, parity number, and breed of the sows from Farm 1 (overall), Farm 1 ’21, and Farm 2.

Variable	Farm 1	Farm 1‘21	Farm 2
RSV1	8.61	8.79	8.9
Day 1 (μ)	30	28	29
RSV2	9.3	9.44 ^1^	9.36 ^1^
Day 2 (μ)	46	45	44
*n*	4165	474	438
Litter Size	13.77	15.26^ 2^	14.24 ^2^
Parity (μ)	3.73	3.25	3.22
Sow breed			
F1	839	264	264
F2	321	-	-
LW	45	29	14

^1^*t*-test, *p* < 0.05, ^2^
*t*-test, *p* < 0.01, *n*: number of cases.

**Table 2 animals-12-01948-t002:** Mean RSV values with days of the scan and litter size (LS) of the sows, grouped in weekly intervals of Day 1. a. Farm 1 (overall), b. Farm 2.

**a.** **Farm 1**			
**Group Day 1** **(*n*)**	**RSV1** **(Mean Day 1)**	**RSV2** **(Mean Day 2)**	**LS**
15–21 (302)	5.15(20)	9.46(37)	14.15
22–28 (1683)	8.33(25)	9.21(42)	13.83
29–35 (1321)	9.5(31)	9.3(48)	13.81
36–42 (584)	9.07(38)	9.46(53)	13.53
≥43 (185)	9.05(48)	9.66 (62)	13.15
**b.** **Farm 2**			
**Group Day 1** **(*n*)**	**RSV1** **(Mean Day 1)**	**RSV2** **(Mean Day 2)**	**LS**
15–21 (24)	5.3(21)	9.67(37)	15.04
22–28 (197)	8.5(25)	9.33(40)	14.28
29–35 (189)	9.7(32)	9.38(47)	14.32
36–42 (22)	9.3(37)	9.09(52)	13.55
≥43 (6)	9.2(44)	9.67(58)	12.16

**Table 3 animals-12-01948-t003:** MAE with s.d. for training, validation and testing.

Metrics	Training	Validation	Testing
MAE s.d.	2.27	2.41	2.58
	1.89	1.98	2.27

**Table 4 animals-12-01948-t004:** Testing MAE with s.d. values for 3-day intervals’ groups in: Day 1, Day 2.

Day 1		Day 2	
Days	MAE (s.d.)	Days	MAE (s.d.)
20–22	2.12	32–37	2.91
	(1.55)		(2.49)
23–25	2.91	38–40	2.54
	(2.66)		(2.16)
26–28	2.46	41–43	2.34
	(2.01)		(1.99)
29–31	2.72	44–46	2.4
	(2.26)		(2.22)
32–34	2.46	47–49	2.62
32–34	(2.26)	2.58 2.27	(2.07)
≥35	2.58	≥50	2.67
	2.26		2.31

## Data Availability

The data presented in this study are available on request from the corresponding author. The data are not publicly available due to farms’ intellectual rights restrictions.

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
