# Peer review of "Evaluation of a Numerical, Real-Time Ultrasound Imaging Model for the Prediction of Litter Size in Pregnant Sows, with Machine Learning"

_animals, 2022, doi:10.3390/ani12151948_

Round 1

Reviewer 1 Report

The paper is overall including interesting data, the syntax can be largely improved. There is incorrect use of english terminology and the sentences can be very hard to understand due to the syntax and grammar used by the authors. 

The abstract needs some re-wording. RTU,DNN must first be spelled out the first time that they are used in the abstract. 

I overall suggest for the authors to have an english speaker colleague or collaborator read through the manuscript, to remove any wrong verbiage (examples: "It is becoming apparent/obvious" has been used few times in the paper and I strongly suggest to find alternative wording; Line 84, 287, 360, 133, 397.. these are just some examples of incorrect english statements).

Line 132: it's missing a space after the end of the sentence.

Line 185: substitute the word "runs"

Author Response

Dear Reviewer 1

Thank you for your valuable comments. 

The changes made on the manuscript, according to your suggestions are:

  1. English syntax and grammar was edited by a UK-resident academic.
  2. The introduction was moderately reviewed to substantiate the use of the references. 
  3. All abbreviations have been spelled-out for their first use in the text.
  4. Line 132: correced
  5. Line 185: substituted

Reviewer 2 Report

Review Report

  • A summary

The study was aimed at evaluation the effectiveness and accuracy of a numerical model based on quantifying real-time ultrasound images of pregnant sows to predict litter size. The approach was based on a field study performed in 2 different commercial pig farms. Deep Neural Network was applied to evaluate the method and its error to predict the litter size. With including the time of diagnosis, this study demonstrated that the lowest mean absolute errors were achieved when the pregnancy diagnosis was performed at days 20-22.

  • Broad comments

The experiments seemed to be carefully done and the paper is thoroughly written.

  • Specific comments

Page 1 line 16: delete an extra “and”.

Page 1 line 22: Use the abbreviation in parentheses (DNN).

Page 1 line 37: Write A.I. in whole since this is used for the first time in the manuscript.

Page 2 line 52:  I would suggest mentioning the type of diagnostic approaches, transabdominal vs. transrectal. The SSE of these two techniques can differ. In addition, the transrectal approach can be classified as a minimal minimally invasive technique and trans-abdominal as non-invasive.

Page 3 line 100: When was the litter size registered in these two farms during the experiment? 

How was cross-fostering practised on two farms during the experiment?

Page 3 line 111: use (AI) after Artificial Insemination

Page 4 lines 124-129:

Page 5 line 178: In your opinion, could some data from previous pregnancies such as previous litter size be included as inputs in this model? And if so, how it can change the results?

Page 6 line 214: fom to from

Page 13 line 376: DL should be written as a whole.

Author Response

Dear Reviewer 2

Thank you for your valuable comments. The ammendments made to the manuscript are:

  1. English language was spell checked.
  2. The methods were moderatelly improved, with explanation on the time of litter size registration and crossfostering.
  3. All specific comments were incorporated in the text.
  4. Page 5, line 178: Data from previous pregnancies is included in the database, as the majority of the sows were recorded for all their parities during the experimental period (7 years). However, the sow id, as an indicator of the tendency of the animal to produce litters of similar size, is considered a very prominent factor to be included in the inputs of the model and your suggestion will be thoroughly considered in the model's future development. Thank you!

Reviewer 3 Report

Line 21 - do not start a sentence with a number.  Please write it out

Line 102 - remove the α symbol and replace with the letter "a"

Please redefine RSV in the body of the paper.  It is only defined in the abstract.

Also, please give a clear description of what RSV is and how it is measured.  A statement is made, but it is not clear what a larger value represents.

Author Response

Dear Reviewer 3

Thank you for the valuable comments. The ammendments on the manuscript that were made accordingly are:

  1. Introduction was reviewed to verify the used litterature.
  2. The description of the methods was enriched, and RSV was further explained in the text
  3. LIne 21: "A number of" added
  4. LIne 102: corrected